# Investigating the Effect of Aggregate Characteristics on the Macroscopic and Microscopic Fracture Mechanisms of Asphalt Concrete at Low-Temperature

**DOI:** 10.3390/ma12172675

**Published:** 2019-08-22

**Authors:** Yinshan Xu, Yingjun Jiang, Jinshun Xue, Jiaolong Ren

**Affiliations:** 1School of Highway, Chang’an University, Xi’an 710064, Shaanxi, China; 2Zhejiang Scientific Research Institute of Transport, Hangzhou 310039, Zhejiang, China; 3Key Laboratory for Road and Bridge Detection and Maintenance Technology of Zhejiang Province, Hangzhou 310039, Zhejiang, China; 4School of Civil Engineering and Architecture, Hubei University of Arts and Science, Xiangyang 441053, Hubei, China; 5School of Civil and Architectural Engineering, Shandong University of Technology, Zibo 255000, Shandong, China

**Keywords:** asphalt concrete, fracture mechanisms, macroscopic and microscopic, low-temperature, fracturing modes, aggregate characteristics, DEM

## Abstract

The low-temperature crack of asphalt concrete is considered to be one of the main deteriorations in asphalt pavements. However, there have been few studies on the composite effects of the aggregate characteristics and fracturing modes on the low-temperature cracking of asphalt concrete. Hence, the edge cracked semi-circular bend tests and the discrete element modeling approaches are combined to investigate the effect of the aggregate contents, aggregate morphological features and aggregate distributions on the fracture behavior of asphalt concrete in different fracturing modes at different temperatures. The results show that the fracture toughness and the crack extended time reduce with the increasing aggregate orientation and flatness and the decreased aggregate content. The effect of aggregate flatness is nonlinear, and its reduction trend grows gradually with the increasing flatness. The total number of failed contacts is reduced with the increasing aggregate orientation and flatness, particularly at 10 °C. The number of failed contacts that occurred in the aggregate–mastic interface in Quasi-Mode II fracturing is slightly higher than that in other fracturing modes. The aggregate distribution in the crack initiation zone greatly influenced the crack resistance, particularly at 10 °C. The research is beneficial to better understand the fracture mechanisms of asphalt concrete at low-temperature.

## 1. Introduction

The low-temperature crack of asphalt concrete is considered to be one of the main deteriorations in asphalt pavements. As the maintenance of the cracked asphalt pavement is cost and time consuming, it is essential to reveal the cracking mechanisms from the view of materials and structures, to mitigate cracking propagation via the whole asphalt overlay [1,2].

According to an investigation performed by Ameri et al. [3] and Aliha et al. [4,5], the cracked asphalt pavement subjected to traffic loading generally experienced combinations of mixed conditioned fracturing modes, including Mode I fracturing, Mode II fracturing, and Mixed-Mode I and II fracturing, depending on the location of the vehicle wheels relative to the crack. Investigating the fracturing behaviour under different fracturing modes is beneficial to better understand the fracture mechanisms. Conversely, asphalt concrete is a combination of coarse aggregates, asphalt mastic and air voids [6], of which approximately 50% of the total volume is coarse aggregates. Obviously, the low-temperature crack resistance of asphalt concrete is influenced by the behavior of all of the coarse aggregates, asphalt mastic, and voids, particularly by the coarse aggregates. The aggregate characteristics, such as the gradation, lithology, morphological features, and distribution, influence the aggregate resistance to the cracking process and, therefore, the performance of asphalt pavements [7,8]. It is important, therefore, to study the fracture performance and crack propagation of asphalt concrete containing different types of aggregate characteristics in different fracturing modes at low temperature, to completely reveal the cracking mechanisms.

The fracture failure of asphalt concrete at low-temperature is complex, because it essentially depends on complex microstructures, temperature and fracturing modes, causing complex fracture patterns consisting of the main crack, crack branching and crack deflection [9,10]. Granting laboratory experiments have been widely adopted to investigate the fracture resistance [4,11] and cracking process [12,13] of asphalt concrete, which analyze the effect of temperatures [1,14,15,16], environmental factors [17,18] and fracturing modes [2,9,14,19], it is difficult to clearly reveal the effect of some important aggregate characteristics (e.g., morphological features and distribution), and require a high-cost and a lot of time for manufacturing samples. To contrast, the numerical simulation method has the advantage of characterizing these complex microstructures, saving research time and reducing experimental samples [20]. Hence, the effect of aggregate characteristics on the low-temperature fracture considered micro-structural details using a homogeneous modeling approach and has been investigated successfully.

Mahmoud et al. [21] revealed the effect of coarse aggregate lithology and gradation on the micro-forces of asphalt concrete during the indirect tensile fracture process using the discrete element method (DEM). Analysis revealed that the asphalt concrete using the aggregate gradation of the porous friction course and the aggregate lithology of the soft limestone experienced the highest micro-forces. Yin et al. [20,22,23] simulated three-point bending tests based on the finite element method (FEM) to investigate the low-temperature cracking process and internal stress of asphalt concrete subject to different aggregate distributions and temperatures in Mode I fracturing. Results showed that the low-temperature cracks of asphalt concrete presented elastic–brittle characteristics when the temperature was lower than 0 °C, and that was significantly changed when the temperature was higher than 5 °C because the effect of the viscoelasticity provided by asphalt mastic was gradually increased. The aggregate distribution locally changed the crack path due to the obstruction of coarse aggregates to the crack propagation. Wang et al. [24] analyzed the relationship of tensile stress, crack initiation locations and fine aggregate mastic damage of asphalt concrete using FEM in Mode I fracturing. It was found that the interface characteristics of coarse aggregate and mastic played an important role in the cracking development. Gao et al. [25] employed DEM modeling technology to reveal the effect of aggregate distribution on the crack distribution of asphalt concrete using the three-point bending test in Mode I fracturing. The study indicated that the cracks trended to propagate inside the matrix or along interfaces between aggregates and matrix with the increasing temperature due to the gradually decreasing fracture resistance of the matrix compared to the aggregates. Ren et al. [26] revealed the effect of micro-aggregate heterogeneity on the cracking process of asphalt concrete. It was found that a large number of micro-cracks initiated very quickly and gradually became stable before the peak load was reached. The cracks continued to propagate and coalesce with newly formed cracks in the cement paste phase. Chen et al. [27] analyzed the effect of material composition on the tensile strength of asphalt concrete at low temperature. Compared to air voids and aggregate modulus, coarse aggregate volume fraction had a more significant effect on the fracture behaviors. Sun and Ren [28] investigated the fracture characteristics of asphalt concrete in Mixed-Loading Mode at low temperature. Results showed that the localized fracture showed the crack path was formed by bridging the initiated interfacial micro-cracks and, meanwhile, nearby cracks became unloaded and closed. The above studies contribute considerably to the understanding of the cracking mechanisms of asphalt concrete.

There are few studies, however, on the effect of fracturing Modes, particularly the Mixed-Mode I and II fracturing and the Mode II fracturing. Hence, it is important for the composite analysis of mixed fracturing Modes, aggregate characteristics and temperatures on the fracture behavior of asphalt concrete, which is an enormous and complex work. The main innovative contribution here is to investigate the coupling effect of the fracturing Modes and aggregate characteristics on the cracking process of asphalt concrete at low temperatures. Based on this, the fracture properties and cracking process of asphalt concrete containing different aggregate contents, aggregate morphological features and aggregate distributions in Mode I fracturing, Quasi-Mode II fracturing and Mixed-Mode I and II fracturing at different temperatures via laboratory tests and numerical simulations are analyzed.

## 2. Materials

An AH-70 base asphalt binder manufactured in Shanxi Province, China, is used, where the physical properties are tested according to the Chinese test specification called the “*Standard test methods of bitumen and bituminous mixtures for highway engineering (JTG E20-2011)*”, which is plotted in Table 1.

The aggregate gradation of the asphalt concrete is selected from the Chinese design specification “*Specifications for design of highway asphalt pavement (JTG D50-2017)*”. The asphalt mastic is obtained by mixing the asphalt binder and the aggregates with diameters smaller than 1.18 mm. The aggregate gradations of the asphalt concrete, asphalt mastic and coarse aggregates are listed in Table 2. As previously mentioned, the asphalt concrete can be treated as a combination of the aggregates whose diameters are larger than 1.18 mm, the asphalt mastic and the air voids [6]. The asphalt mastic can be considered as a combination of the aggregates with diameters smaller than 1.18 mm and the asphalt binder, in other words. Figure 1 plots the preparation process of the asphalt mastic samples and asphalt concrete samples, which can be beneficial to show the relationship of the asphalt concrete, asphalt mastic and asphalt binder. The binder mass ratio and aggregate gradation of the asphalt mastic samples are the same as that of the asphalt mastics in the asphalt concrete samples.

The asphalt mastic samples and asphalt concrete samples, of which the dimension are all *φ* 100 mm × *h* 150 mm, are compacted in a Superpave gyratory compactor, according to the Chinese test specification called the “*Standard Test Methods of Bitumen and Bituminous Mixtures for Highway Engineering (JTG E20-2011)*”. Three types of asphalt concrete composed of different coarse aggregate contents are analyzed, and the compositions are shown in Table 3. It should be noted that the gradation of the coarse aggregates and asphalt mastic is the same in different asphalt concretes.

## 3. Laboratory Test and Discussion

### 3.1. Laboratory Test

Due to the advantages of introducing various fracturing Modes (Mode I fracturing, Mixed-Mode I and II fracturing and Quasi-Mode II fracturing) and preparing samples conveniently [9,11,30], the edge cracked semi-circular bend (SCB) test has been widely adopted to investigate the cracking behavior of asphalt concrete in recent years. Figure 2 shows the preparation process of the SCB sample. One cylindrical sample of asphalt concrete can prepare six SCB samples. Figure 3 displays the prepared SCB samples and the SCB testing apparatus. The sample preparation and sample size of the SCB tests is the same as that used in the authors’ previous research [31]. Prior to conducting the SCB tests, the prepared samples are maintained inside a freezer for more than 4 h at a constant temperature. The frozen samples are then placed inside the loading apparatus in an environmental chamber based on a UTM servo-hydraulic testing system.

Regarding the SCB tests, the different fracturing Modes can be realized via changing the geometry factors of the SCB samples, which were verified previously by Ayatollahi et al. [32] and Ameri et al. [9]. The mixity parameter *M^e^* calculated using Equation (1) is to determine the fracturing Modes (*M^e^* = 1 (Mode I) to *M^e^* = 0 (Mode II)), which could be obtained from the study of Ameri et al. [9].(1)$$M^{e}=\frac{2}{\pi }\tan^{-1}(\frac{K_{I}}{K_{II}})$$
where *K_I_* and *K_II_* = the Mode I and Mode II stress intensity factors, respectively, calculated via Equation (2), as follows:(2)$$K_{I}=\frac{P}{2Rt}\sqrt{\pi a}Y_{I}(\frac{a}{R},\frac{S1}{R},\frac{S2}{R}),\;K_{II}=\frac{P}{2Rt}\sqrt{\pi a}Y_{II}(\frac{a}{R},\frac{S1}{R},\frac{S2}{R})$$

Three types of coarse aggregate content (No.1, No.2 and No.3, as shown in Table 2), three temperature grades (−6 °C, 0 °C and 10 °C), and three types of fracturing Modes (Mode I, Quasi-Mode II and Mixed-Mode I and II) are selected to implement SCB tests. The loading conditions corresponding to different fracturing modes are displayed in Figure 4, which can be obtained from the study of Ameri et al. [9]. The loading rate is 1 mm/min in all tests. Six samples are tested successfully for each loading condition and concrete composition.

### 3.2. Test Results and Discussion

The fracture toughness and the crack extended time are employed to describe the fracture properties of the asphalt concrete. The fracture toughness *K* is calculated via Equation (1). The crack extended time is equal to the time at which peak load occurred. The crack length that occurred in the aggregates, mastics and aggregate–mastic interface is used to analyze the crack distribution. The test results are shown in Table 4 and Table 5 where *T* is temperature, AV is the average value, and SD is the standard deviation.(3)$$K=\sqrt{K_{I}^{2}+K_{II}^{2}}$$
where, *K_I_* and *K_II_* = the stress intensity factors in Mode I and II, *Y_I_* and *Y_II_* = the geometry factors (could be obtained from the study of Ameri et al. [9]) corresponding to Mode I and II, *P* = the applied load, *R* = the sample radius, *t* = the sample thickness, and *a* = the edge vertical crack length.

As previously mentioned, the coarse aggregate content is the highest in the No.1 asphalt concrete and the lowest in the No.3 asphalt concrete (Table 2). The following observations can be made according to the test results shown in Table 4 and Table 5:(1)The fracture toughness and the crack extended time, at the same temperature, are increased with the increasing coarse aggregate content. The previous works [27,33,34] indicate that there is the optimal percentage of the aggregates in asphalt concrete. When the aggregate content is lower than the optimal percentage, the fracture resistance of asphalt concrete is negatively affected with the decreased aggregate content. This is because the aggregate skeleton structure is weakened gradually with the decreasing coarse aggregate content, and the fracture resistance drops accordingly. The cracks will occur more easily and extend faster.(2)The total crack length in the asphalt concrete, the crack length in the aggregates, and the crack length in the aggregate/mastic interface diminishes with the decreasing coarse aggregate content, while the crack length in the mastic increases. This is because the stiffness of the coarse aggregate is much higher than that of the asphalt mastic, and the crack can be obstructed by the coarse aggregate ahead of its tip. Accompanying the decrease in the coarse aggregate content, cracks pass the mastic directly and easily, and do not have to bypass the coarse aggregate. The crack complexity is also weakened.(3)The effect of the coarse aggregate content on the crack distribution in Quasi-Mode II is more significant than that in other fracturing modes, particularly at 10 °C, showing that the coarse aggregate content influences the crack resistance of asphalt concrete more sensitively in shear loading conditions and at a higher temperature.(4)The crack distribution has less difference between −6 °C and 0 °C, and it is different from that at 10 °C. Moreover, according to the previous studies [1,23], the low-temperature cracks of asphalt concrete present the elastic—brittle characteristics when the temperature is lower than 0 °C, which will be changed significantly when the temperature is higher than 5 °C because the effect of viscoelasticity provided by asphalt mastic is gradually increased. Hence, −6 °C and 10 °C are selected in the following study.

## 4. DEM Simulation and Discussion

### 4.1. Simulation Preparation

A 2D-DEM modeling technology proposed in the authors’ previous studies [26,28] using the software *PFC* is used to build the SCB numerical sample, as shown in Figure 5. The numerical sample contained coarse aggregate elements (red region) with the diameter larger than 1.18 mm, asphalt mastic elements (black region) and air void elements (white region). Both the coarse aggregate elements and asphalt mastic elements are treated as densely assembled balls with an equal radius (0.5 mm), and there is no ball in the void elements. Four types of contacts between the balls are used to represent the following four different interactions: contacts within aggregates, between adjacent aggregates, between aggregates and mastics, and within mastics. A series of contact constitutive models established in the authors’ previous work [26,35] are given to the four contacts to describe the elastic and viscoelastic force-displacement relationship of each contact. The values of model parameters are suggested by Sun and Ren et al., (2018). The details of the calibration process can also be referred to the previous work (Sun and Ren et al., 2018). The modeling procedure of the SCB numerical sample is shown in Table 6.

To ensure the reliability of the simulations, parallel testing is implemented in numerical simulations, just as those used in the experimental tests. During the experimental parallel tests, the asphalt concrete samples usually have the same raw materials and material composition, and random material distribution. Accordingly, in the numerical parallel tests, the numerical models have the same material parameters and material composition, and random material distribution. As previously mentioned, for each loading condition and concrete composition, six samples are tested in experimental tests. Similarly, six numerical models are simulated successfully for each loading condition and concrete composition. The average errors between the DEM simulation results and SCB test data are listed in Table 7.

Table 7 indicates a reasonable comparison between the experimental data and simulation results. The average error of the different indexes is lower than 6.66%. However, the microstructure is idealized as a 2D status, while in reality, the 3D effect will influence the fracture process, which is a limitation of this study. Nevertheless, due to a high computing time and complexity in conducting 3D models, it is not necessary to build a mass of models. Many current studies for the fracture analysis of asphalt concrete have focused on performing 2D simulations and have achieved satisfactory results [20,21,22,36,37]. Hence, the current DEM approach can provide a useful tool to investigate the cracking characteristics.

### 4.2. Simulation Plan

Aggregate morphological features and aggregate distribution are difficult to replicate accurately in lab tests. Numerical simulations thus provide a powerful tool to analyze these characteristics.

Three types of aggregate orientations (0°, 45° and 90°), three types of aggregate flatnesses (1.0, 1.1 and 1.2), and three types of aggregate distributions are selected to analyze the effect of the aggregate characteristics on the fracture properties and crack propagation. Following the authors’ previous study [28], the crack propagation zone of the SCB models is determined according to the statistics of the test results, as shown in Figure 6, and the difference of Zone I and Zone II is displayed in Table 8. Figure 7, Figure 8 and Figure 9 display examples of numerical models using different aggregate characteristics. The void characteristics are constant in Figure 7 and Figure 8. Moreover, it should be explained that the following simplification is employed in some asphalt concrete models:

(a) To control the aggregate orientation more conveniently, the aggregates are hexagonal in the 0° model, 45° model and 90° model.

(b) The void distribution is difficult to keep constant in the models using different aggregate distributions, because the number of aggregate elements is much higher than that of the void elements. Hence, there is no void element in the different aggregate distribution models to avoid the interference of void distribution.

### 4.3. Simulation Results and Discussion

#### 4.3.1. Fracture Properties

Figure 10 and Figure 11 plot the fracture toughness and the crack extended time of different numerical models, respectively.

According to Figure 10 and Figure 11, the performance ratio of the different models is listed in Table 9 and Table 10. Thereinto, *F*_0°_ (*C*_0°_), *F*_45°_ (*C*_45°_), *F*_90°_ (*C*_90°_), *F*_1.0_ (*C*_1.0_), *F*_1.1_ (*C*_1.1_), *F*_1.2_ (*C*_1.2_), *F*_1#_ (*C*_1#_), *F*_2#_ (*C*_2#_) and *F*_3#_ (*C*_3#_) are the average fracture toughnesses (the crack extended time) in the 0° orientation model, 45° orientation model, 90° orientation model, 1.0 flatness model, 1.1 flatness model, 1.2 flatness model, 1# distribution model, 2# distribution model and 3# distribution model, respectively.

Shown in Table 9 and Table 10, the following observations can be made:(1)The fracture toughness and the crack extended time reduced with the increase of aggregate orientation and aggregate flatness at the same temperature, indicating that the increase of the orientation angle and flatness bring about a negative effect on the fracture properties. This is due to the following: (a) the cracking speed is faster with the increasing aggregate orientation because the crack path that bypasses the coarse aggregates is shortened accordingly, and (b) the crack resistance of the coarse aggregates is reduced with the increase of the aggregate flatness because the aggregates become thinner. Moreover, the effect of the aggregate flatness on the fracture properties is more significant than that of the aggregate orientation.(2)The effect of the aggregate orientation on the fracture properties is approximately linear, while that of the aggregate flatness is nonlinear. The effect degree elevates gradually with the increasing flatness. The reason is that the decay rate of the crack resistance of aggregates is increased with the decrease of the aggregate thickness. Additionally, the effect of the aggregate characteristics at 10 °C is more significant than that at −6 °C, particularly for the aggregate flatness, due to the difference between aggregates and mastics at different temperatures.(3)The effect of the aggregate characteristics on the fracture properties in Quasi-Mode II fracturing is the most significant, particularly at 10 °C. It also illustrates that the effect of the aggregate characteristics on the cracking resistance is more sensitive under the shear loading conditions, due to the loading sensitivity of asphalt concrete under the shear loading conditions. Additionally, it can be found that: *F*_2#_ > *F*_1#_ > *F*_3#_ and *C*_2#_ > *C*_1#_ > *C*_3#_, which shows an obvious correlation between the aggregate distribution in Zone I and the fracture properties.

#### 4.3.2. Crack Propagation

Found in the DEM numerical models, the distribution of the failed contacts implied the potential for crack propagation [27,38]. Figure 12 and Figure 13 plot two types of distribution of failed contacts, namely: (a) failed contacts that occurred in mastic, aggregates and aggregate/mastic interfaces, and (b) failed contacts that occurred along the main crack path and away from the main crack path. The failed contacts that occurred away from the main crack path indicate the potential for crack branching and redistribution.

Wang et al. [24] and Gao et al. [25], as previously mentioned, indicate that the cracks tend to propagate inside the matrix or along interfaces between aggregates and matrix with increasing temperature, which also is proven in this study. Figure 13 and Figure 14 show the total number of failed contacts at 10 °C is larger than that at −6 °C, particularly in Quasi-Mode II. These increased failed contacts at 10 °C mainly appear in the mastic and interface and away from the main crack path. It can be speculated the reason is that the difference between aggregate stiffness and mastic stiffness increases with the rising temperature, according to the analysis of Yin et al. [24] and Gao et al. [25]. Looking at −6 °C, the difference between aggregate stiffness and mastic stiffness is relatively small—the cracks are intended to pass through the aggregates and mastics directly. The number of failed contacts that occurred away from the main crack path is very limited, and the number of failed contacts that occurred in the aggregates, mastics and interfaces has little difference. Regarding 10 °C, the cracks are intended to pass through the mastics and interfaces and bypass the aggregates due to the increasing stiffness difference between aggregates and mastics. Due to the complexity of the aggregate distribution, the cracks are more complex (the crack branching and redistribution can be observed), and the failed contacts easily emerge away from the main crack path. Additionally, the invisible failed contact in *PFC* is defined as the contact force equals zero but the balls are not separated, which indicate untransmitted loads. The invisible failed contacts, in other words, have not formed the visible crack, e.g. the main crack. Hence, it should be explained that most of the failed contacts that occurred in the interface and away from the main crack path are invisible failed contacts at 10 °C.

Figure 12a and Figure 13a show the total number of failed contacts is reduced with the increasing aggregate orientation, particularly at 10 °C. The reduced failed contacts at 10 °C mainly occurred in the interface and along the main crack path. This is because the length of the crack path that bypasses the aggregates is reduced with the increasing aggregate orientation. The change law of failed contacts in the numerical models using different aggregate orientation is similar in the different fracturing modes.

Figure 12b and Figure 13b show the total number of failed contacts is reduced with the increasing aggregate flatness. Occurring at −6 °C, there are only a few failed contacts, but it is worth noting that this reduction of failed contacts mainly appears in aggregates. Regarding 10 °C, the number of failed contacts has a greater variation with the change of aggregate flatness, for which the effect trend increases gradually with the increasing flatness. This is because the aggregates grow thinner with the increasing aggregate flatness, and the aggregate crack resistance is reduced accordingly. Hence, although the difference of aggregate stiffness and mastic stiffness increases with the rising temperature, the cracks at 10 °C still are inclined to pass through the aggregates directly, but not to bypass them. Thus, it also can be found that the failed contacts begin to appear in aggregate elements with the increasing aggregate flatness. The number of failed contacts that occurred in the interface and along the main crack path is reduced with the increasing aggregate flatness because the crack length and complexity are weakened.

Yin et al. [20,22,23] indicate that the aggregate distribution can locally change the crack path but only has a slight influence on the overall cracking direction. Their studies, however, are only implemented in Mode I fracturing and do not apply to the irregular shape of coarse aggregates in the numerical models. Hence, these current findings have a few differences with the existing results. Figure 12c and Figure 13c show there is little difference among the numerical models using different aggregate distributions at −6 °C. It is worth noting, however, that the number of failed contacts that occurred in the interface in Quasi-Mode II is slightly higher than that in other fracturing modes. Considering 10 °C, the total number of failed contacts in 3# model is the least. This is because the aggregate content of the 1# model and 2# model in Zone I is higher than that of 3# model, thus the cracks bypass the aggregates and the path is lengthened. It shows that the aggregate distribution in Zone I has a greater influence on the crack propagation. Additionally, the difference between the total number of failed contacts in the 1# model and 2# model is not obvious in Mode I, but in Quasi-Mode II and Mixed-Mode I and II, the total number of failed contacts of the 2# model is slightly higher than that of the 1# model. The increased failed contacts of the 2# model mainly occur in the interface and along the main crack path. It indicates that the aggregate characteristics in Zone II also affect the crack propagation in the shear loading condition.

The difference of the three aggregate distribution models is shown in Figure 14, of which the fracture morphology after fracture failure is plotted in Figure 15.

Shown in Figure 14, the aggregate distribution at the crack propagation zone in the different models has the following differences:

(1) Occurring in Zone I, there is a large aggregate in the center of the pre-cut crack in the 1# model and 2# model. Found in the 3# model, the larger aggregate exists on the right side of the pre-cut crack, as shown in the white circles in Figure 14.

(2) Viewing Zone II, some large aggregates are located in the center region in the 2# model and 3# model, while in the 1# model, the diameters of the aggregates in the center part are relatively small, as shown in the green circles in Figure 14.

Combining Table 8, Figure 14 and Figure 15a,c,e, the cracks of 1# model and 2# model in Zone I at −6 °C pass through the coarse aggregates directly, and that of 3# model are inclined cross one side of the aggregates. The aggregate characteristics in Zone I mainly influence the crack path. To demonstrate: (1) the aggregate content of 1# model in Zone II has been reduced, but the crack still passes through the aggregates directly, and (2) the aggregate content of 3# model in Zone II has been increased, but the crack path of 3# model in Zone II is also inclined to bypass the aggregates rather than to pass through the aggregates. The above implies the importance of aggregate characteristics in Zone I for the cracking process.

Figure 15b,d,f, shows the cracks in 1# model and 2# model are more complex than that in 3# model at 10 °C, particularly in Quasi-Mode II, because the cracks tend to bypass the large aggregates near the pre-cut notch in Zone I. Additionally, it is worth noting that 1# model in mixed mode I and II at 10 °C is cracked from the right support. This is because the crack resistance of the mastics and small aggregates near the right support is much lower than that of the large aggregates in Zone I near the pre-cut notch. This phenomenon also can be observed in a few experiments at 10 °C, accidentally, the main reason being the difference of aggregate distribution between near the pre-cut notch and the support. Moreover, it should be explained that the crack that occurred near the support is a special case in a very few numerical simulations and laboratory tests. The sample with this phenomenon both in numerical simulations and laboratory tests is considered as the invalid sample and must be retested.

## 5. Conclusions

The composite effects of the aggregate characteristics and fracturing modes on the cracking behavior of asphalt concrete are investigated via SCB laboratory tests and DEM numerical simulations at two typical temperatures (−6 °C and 10 °C).

Found at −6 °C, the fracture of asphalt concrete presents elastic–brittle characteristics. The cracks are intended to pass through the coarse aggregates and asphalt mastics rapidly and directly. The aggregate characteristics only have a little influence on the fracture behavior of asphalt concrete, except for the aggregate flatness. The fracture toughness of asphalt concrete reduces clearly with the increasing aggregate flatness. The reason is that the crack resistance of the coarse aggregates is weakened as the aggregates become thinner with the increasing aggregate flatness.

Observed at 10 °C, the cracks of asphalt concrete are intended to pass through the asphalt mastics and aggregate–asphalt interfaces and bypass the aggregates due to the increasing stiffness difference between concrete aggregates and asphalt mastics. Moreover, the crack resistance and the crack length are reduced with the increasing aggregate orientation angle because the cracks that bypass the coarse aggregates are shortened with the increasing orientation angle. Thus, at −6 °C, the crack resistance reduces with the increasing aggregate flatness, and the cracks that occurred in aggregates can be observed at 10 °C.

The fracture toughness is the lowest in Mixed-Mode I and II fracturing, and the crack time is the highest in the Quasi-Mode II fracturing. The chance of cracking behavior in different fracturing modes at 10 °C is more significant than that −6 °C. Compared to other fracturing modes, the cracks are more complex, and the effects of aggregate characteristics are more significant in Quasi-Mode II fracturing, particularly for the aggregate flatness. It shows that the crack resistance of asphalt concrete is more sensitive in shear loading conditions and at a higher temperature.

The aggregate distribution in the initial zone of the crack propagation plays an important role in the crack development, particularly the distribution of coarse aggregates with a larger diameter, which can influence the crack path comprehensively.

The analyses of the crack behavior of asphalt concrete caused by temperature changes and environmental effects are useful to reveal the fracture mechanisms further, and are being carried out and will be illustrated in the future.

## Figures and Tables

**Figure 1 materials-12-02675-f001:**
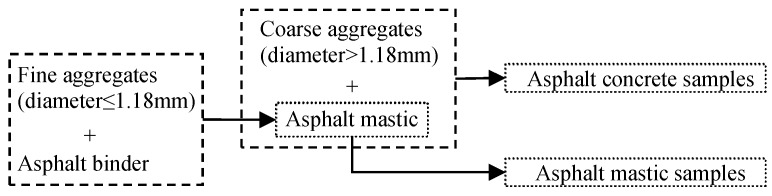
Preparation process of the asphalt concrete sample and asphalt mastic sample.

**Figure 2 materials-12-02675-f002:**
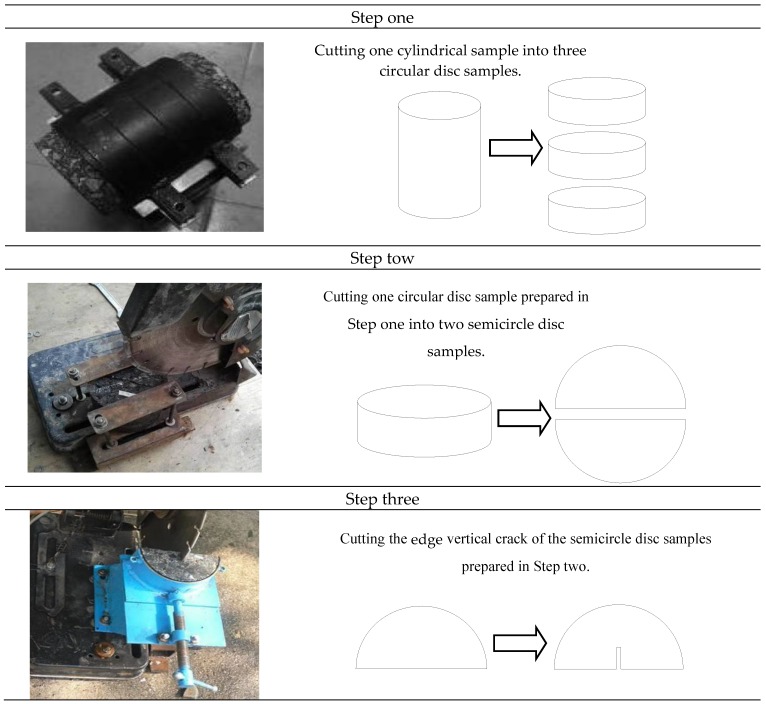
Preparation process of the SCB sample.

**Figure 3 materials-12-02675-f003:**
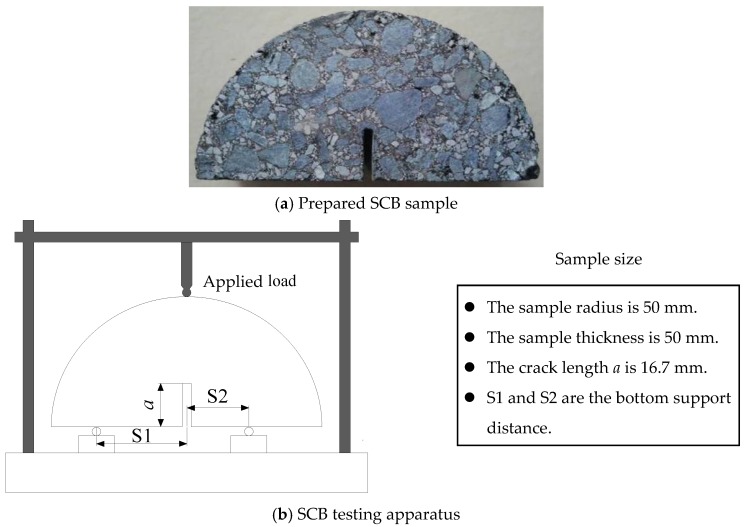
Prepared SCB sample and SCB testing apparatus. (**a**) Prepared SCB sample; (**b**) SCB testing apparatus.

**Figure 4 materials-12-02675-f004:**
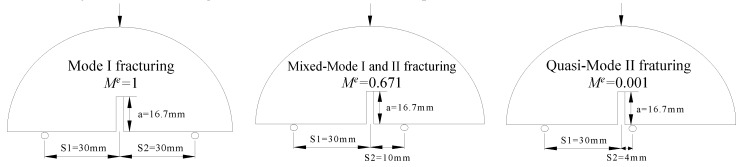
Loading conditions of SCB tests.

**Figure 5 materials-12-02675-f005:**
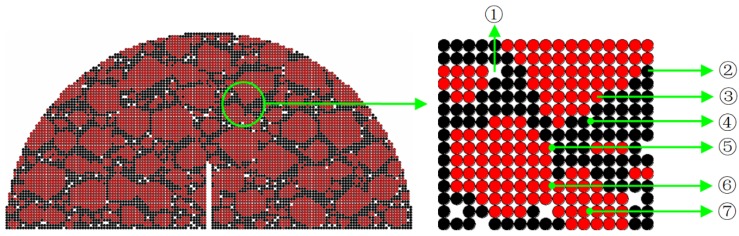
An example of an SCB numerical sample. ①: Air void; ②: Asphalt mastic ball; ③: Aggregate ball; ④: Contacts within the asphalt mastic; ⑤: Contacts between the aggregate and asphalt mastic; ⑥: Contacts between adjacent aggregates; ⑦: Contacts within the aggregate.

**Figure 6 materials-12-02675-f006:**
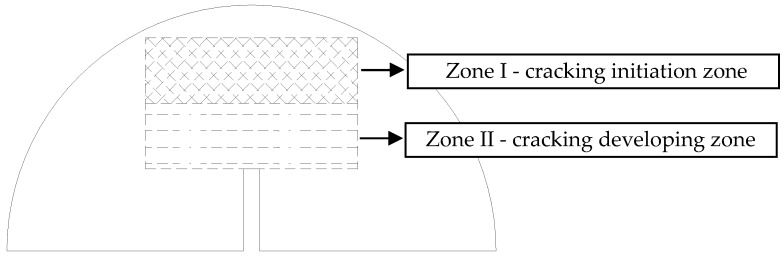
Crack propagation region of DEM model.

**Figure 7 materials-12-02675-f007:**
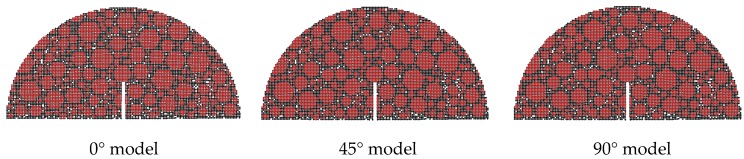
Examples of numerical models using different aggregate orientations.

**Figure 8 materials-12-02675-f008:**
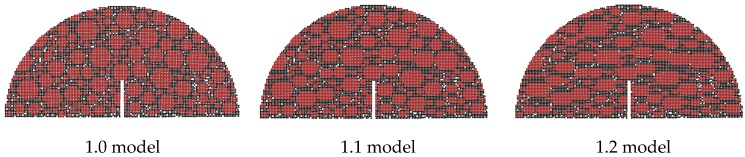
Examples of numerical models using different aggregate flatnesses.

**Figure 9 materials-12-02675-f009:**

Examples of numerical models using different aggregate distributions.

**Figure 10 materials-12-02675-f010:**
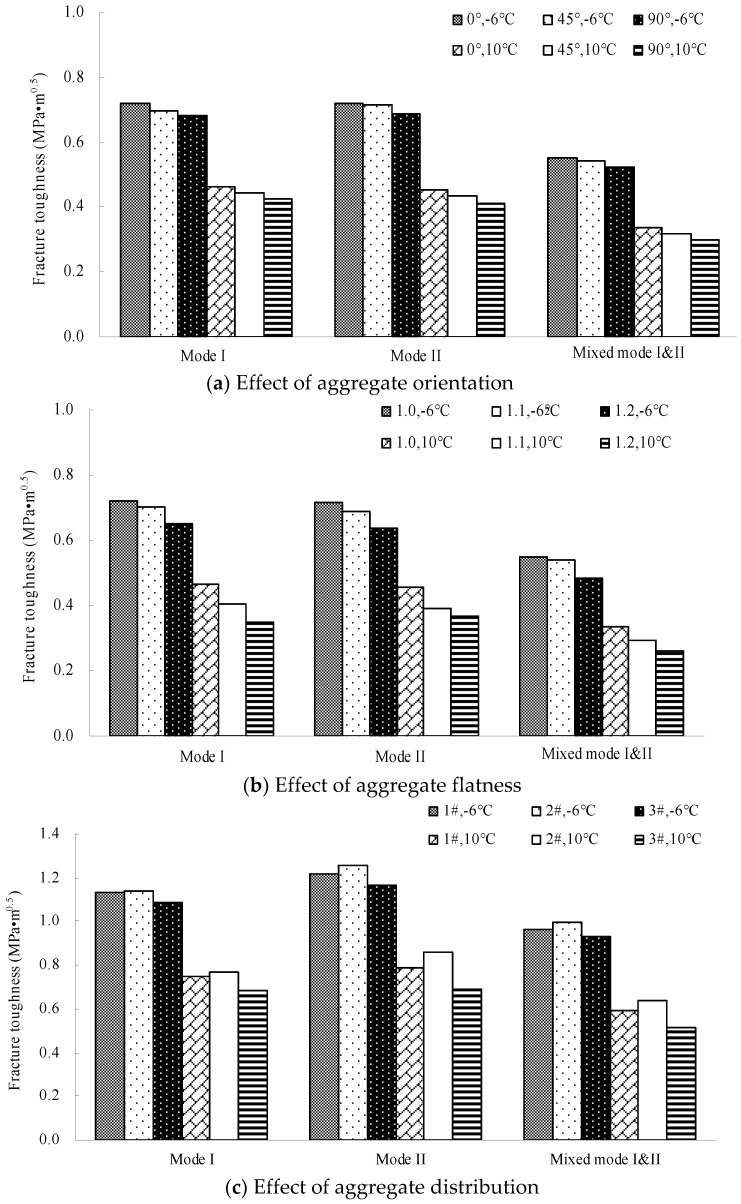
Fracture toughness obtained from different numerical models. (**a**) Effect of aggregate orientation; (**b**) Effect of aggregate flatness; (**c**) Effect of aggregate distribution.

**Figure 11 materials-12-02675-f011:**
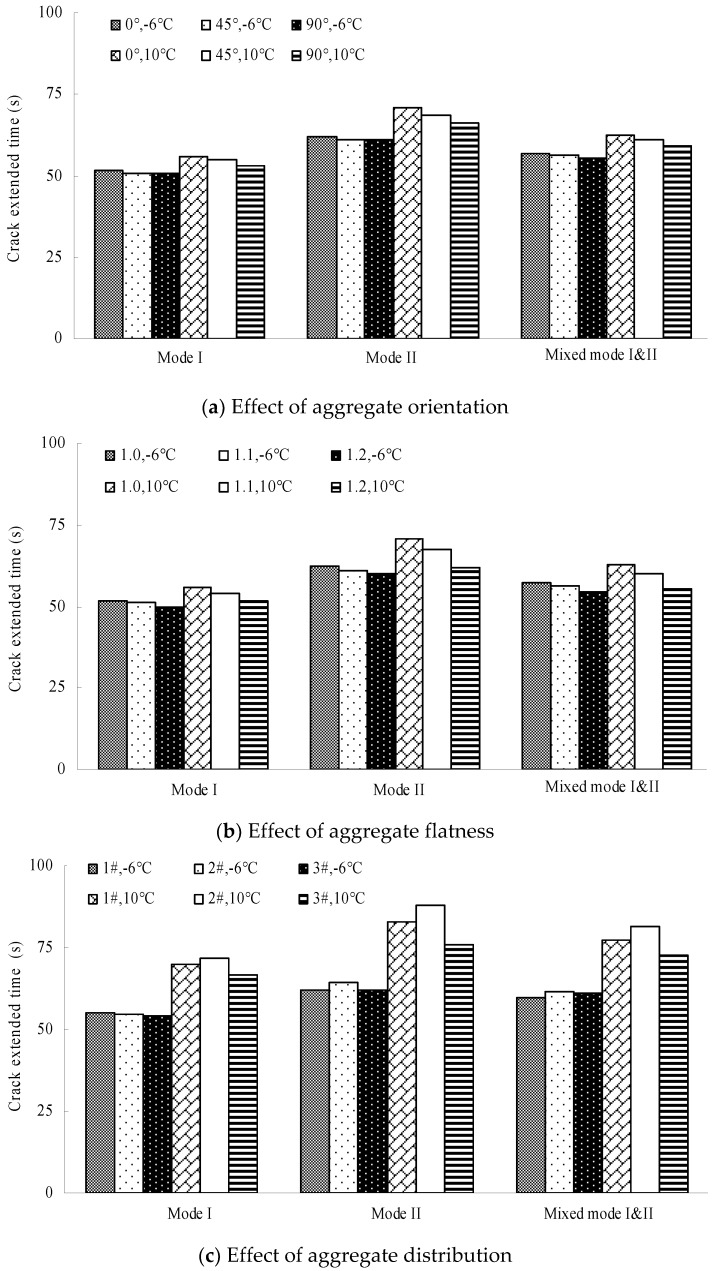
Crack extended time obtained from different numerical models.; (**a**) Effect of aggregate orientation; (**b**) Effect of aggregate flatness; (**c**) Effect of aggregate distribution.

**Figure 12 materials-12-02675-f012:**
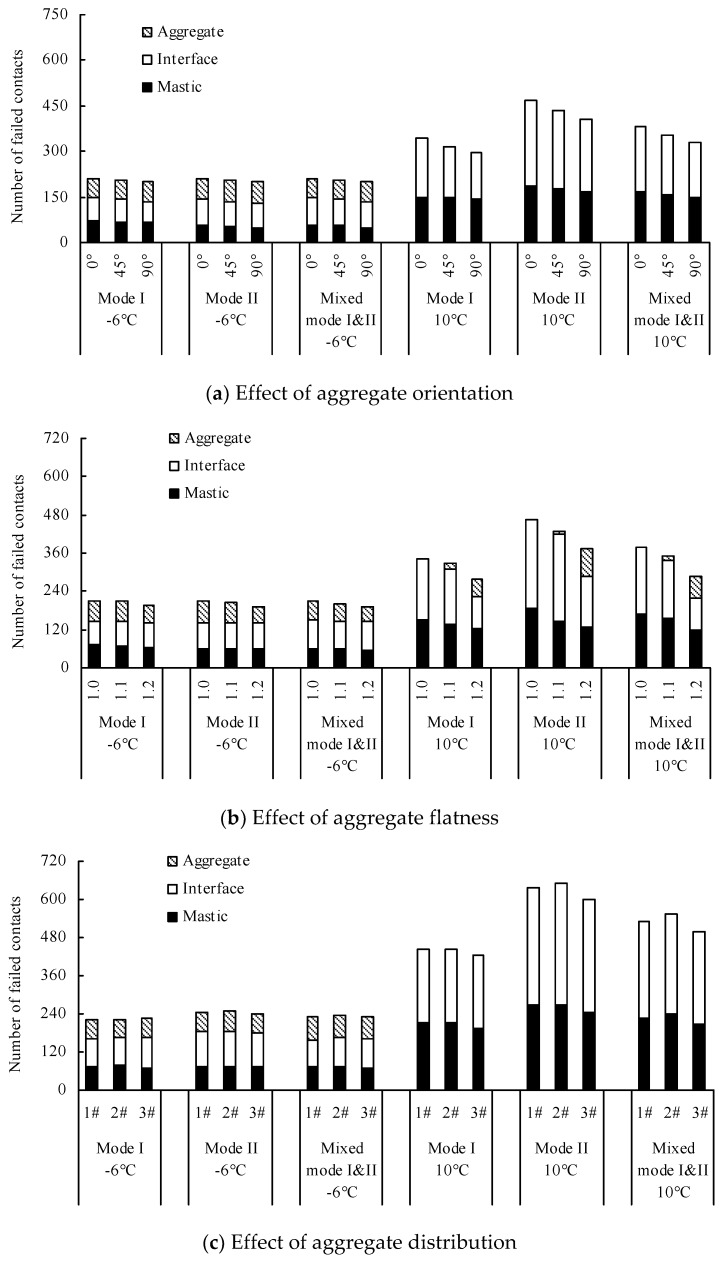
An accumulative histogram of the number of failed contacts that occurred in the mastic, aggregate and mastic–aggregate interface. (**a**) Effect of aggregate orientation; (**b**) Effect of aggregate flatness; (**c**) Effect of aggregate distribution.

**Figure 13 materials-12-02675-f013:**
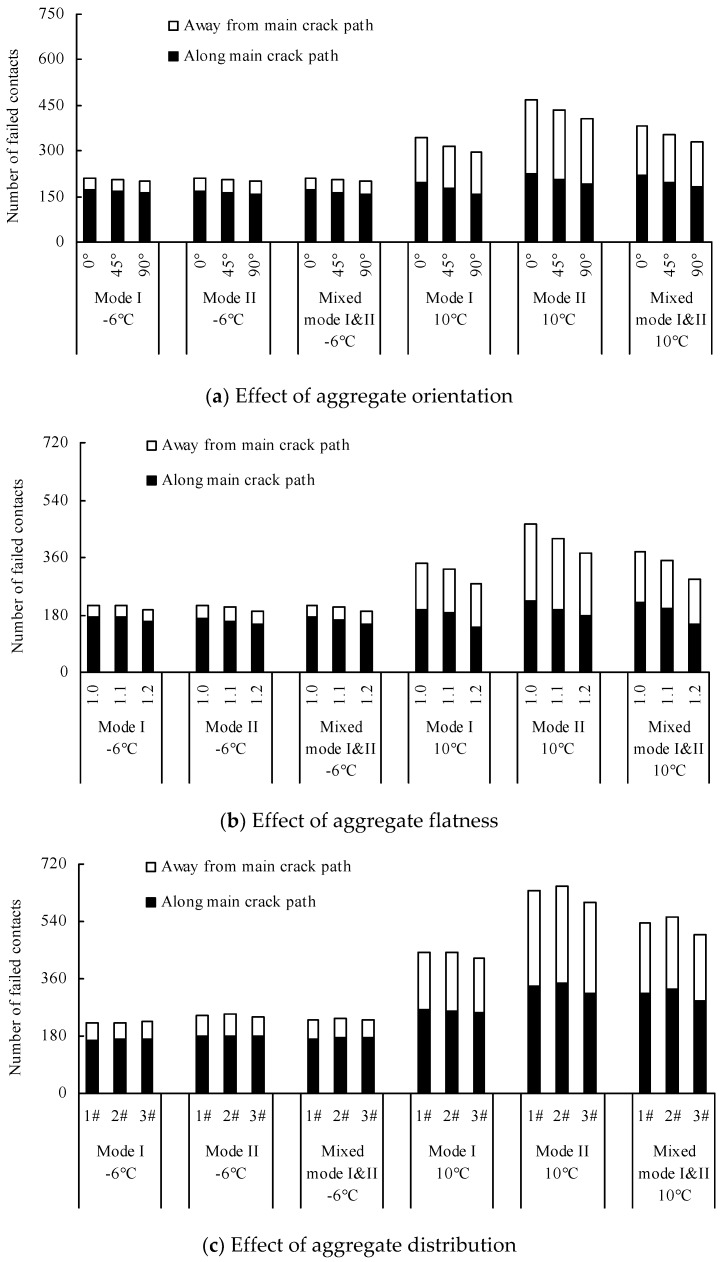
An accumulative histogram of the number of failed contacts that occurred away from the main crack path and along the main crack path: (**a**) Effect of aggregate orientation; (**b**) Effect of aggregate flatness; (**c**) Effect of aggregate distribution.

**Figure 14 materials-12-02675-f014:**
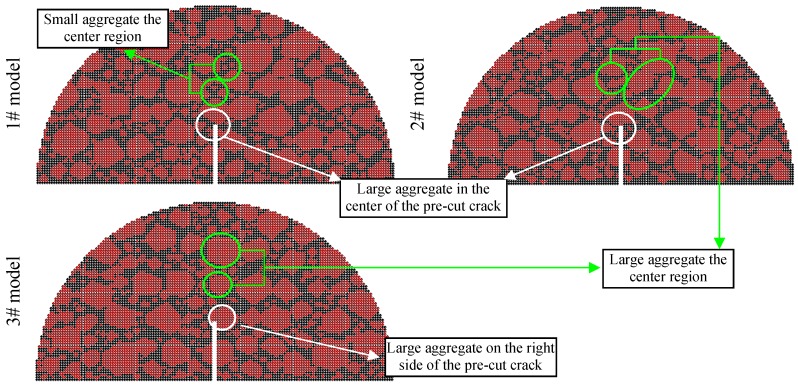
DEM model using different aggregate distribution.

**Figure 15 materials-12-02675-f015:**
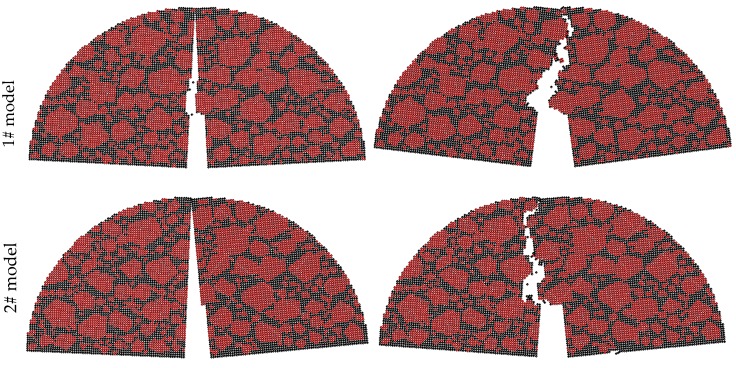
Fracture diagram of different numerical models. (**a**) Mode I at −6 °C; (**b**) Mode I at 10 °C; (**c**) Mixed-Mode I and II at −6 °C; (**d**) Mixed-Mode I and II at 10 °C; (**e**) Quasi-Mode II at −6 °C; (**f**) Quasi-Mode II at 10 °C.

**Table 1 materials-12-02675-t001:** Physical properties of the asphalt binder.

25 °C Penetration (0.1 mm)	Softening Point (°C)	10 °C Ductility (cm)	15 °C Ductility (cm)	60 °C Viscosity (Pa.s)
64.5	55.5	32.1	>100	299

**Table 2 materials-12-02675-t002:** Gradation of the asphalt concrete, asphalt mastic, and coarse aggregate.

Materials	Pass Percent (%) through the Following Sieve Size									
	16 mm	13.2 mm	9.5 mm	4.75 mm	2.36 mm	1.18 mm	0.6 mm	0.3 mm	0.15 mm	0.075 mm
Asphalt concrete	100	87	70	40	28	24	18	11	8	7
Asphalt mastic	--	--	--	--	--	100	75	46	33	29
Coarse aggregate	100	83	61	22	5	--	--	--	--	--

**Table 3 materials-12-02675-t003:** Composition of asphalt concrete ^a^.

No.	$\frac{M_{CA}}{M_{AC}}$	$\frac{V_{CA}}{V_{AC}}$	$\frac{M_{AB}}{M_{AC}}$	$\frac{M_{AB}}{M_{AM}}$	$\frac{V_{AB}}{V_{AC}}$	$\frac{V_{AB}}{V_{AM}}$	$\frac{M_{AV}}{M_{AC}}$	$\frac{M_{AV}}{M_{AM}}$	$\frac{V_{AB}}{V_{AC}}$	$\frac{V_{AB}}{V_{AM}}$	$\frac{M_{AM}}{M_{AC}}$	$\frac{V_{AM}}{V_{AC}}$
No.1	50.5%	38.5%	8.1%	16.4%	19.7%	34.2%	0%	0% ^b^	4%	0% ^b^	49.5%	57.5%
No.2	61.6%	51.4%	6.3%		15.3%						38.4%	44.6%
No.3	72.6%	64.2%	4.5%		10.9%						27.4%	31.8%

^a^*M* is the mass, *V* is the volume; CA is the coarse aggregates; AC is the asphalt concrete; AB is the asphalt binder; AM is the asphalt mastic; AV is the air void. ^b^ The porosity of the asphalt mastic can be negligible due to the high fluidity [29].

**Table 4 materials-12-02675-t004:** SCB test results of the physical and mechanical properties.

*T* (°C)	Aggregate Content	Fracturing Modes	Fracture Toughness (MP·m^0.5^)		Peak Load (kN)		Crack Extended Time (s)	
			AV	SD	AV	SD	AV	SD
−6	No.1	Mode I	0.681	0.101	4.51	0.38	50.0	2.11
		Quasi-Mode II	0.664	0.066	6.71	0.62	61.2	3.07
		Mixed-Mode	0.557	0.046	8.46	0.97	54.9	1.67
	No.2	Mode I	0.808	0.082	5.35	0.21	51.2	3.55
		Quasi-Mode II	0.823	0.099	8.31	0.77	62.8	5.01
		Mixed-Mode	0.667	0.021	10.14	1.12	57.0	2.11
	No.3	Mode I	0.893	0.086	5.91	0.57	52.5	3.36
		Quasi-Mode II	0.898	0.148	9.07	1.49	66.4	3.90
		Mixed-Mode	0.735	0.022	11.17	0.33	59.3	3.08
0	No.1	Mode I	0.555	0.111	3.67	0.76	52.3	2.77
		Quasi-Mode II	0.548	0.067	5.53	0.26	64.5	3.16
		Mixed-Mode	0.451	0.012	6.86	0.81	57.1	1.89
	No.2	Mode I	0.661	0.051	4.37	0.44	54.3	3.33
		Quasi-Mode II	0.667	0.034	6.73	0.50	67.3	1.77
		Mixed-Mode	0.529	0.048	8.04	0.35	59.3	2.91
	No.3	Mode I	0.721	0.068	4.77	0.71	56.7	3.97
		Quasi-Mode II	0.736	0.011	7.44	0.62	73.8	4.01
		Mixed-Mode	0.595	0.099	9.05	0.11	63.4	3.88
10	No.1	Mode I	0.409	0.110	2.71	0.27	56.1	2.22
		Quasi-Mode II	0.396	0.067	4.00	0.34	70.1	4.84
		Mixed-Mode	0.311	0.089	4.73	0.51	60.6	3.77
	No.2	Mode I	0.508	0.034	3.36	0.44	58.0	3.92
		Mode II	0.500	0.051	5.05	0.30	72.1	5.81
		Mixed-Mode	0.393	0.022	5.98	0.21	62.3	4.11
	No.3	Mode I	0.604	0.029	4.00	0.51	60.7	2.19
		Quasi-Mode II	0.616	0.031	6.22	0.46	77.8	4.99
		Mixed-Mode	0.473	0.055	7.20	0.49	66.4	3.60

**Table 5 materials-12-02675-t005:** SCB test results of the crack length.

*T* (°C)	Aggregate Content	Fracturing Modes	Crack Length (mm) That Occurred in Aggregate		Crack Length (mm) That Occurred in Mastic		Crack Length (mm) That Occurred in Interface	
			AVG	SD	AVG	SD	AVG	SD
−6	No.1	Mode I	6.0	0.81	13.9	1.09	10.7	1.07
		Quasi mode II	3.7	0.11	13.5	2.01	12.0	2.11
		Mixed mode	5.4	0.62	12.2	0.97	13.1	1.02
	No.2	Mode I	8.2	1.10	12.6	1.23	12.4	1.68
		Quasi mode II	5.8	0.76	12.9	2.09	13.8	1.31
		Mixed mode	7.3	0.81	11.4	1.62	14.7	1.99
	No.3	Mode I	9.1	1.31	12.0	1.79	13.8	2.12
		Quasi mode II	8.5	1.78	12.4	3.07	15.1	2.88
		Mixed mode	8.9	0.90	10.9	1.02	16.3	1.69
0	No.1	Mode I	4.8	0.38	15.7	2.89	10.9	1.26
		Quasi mode II	3.1	0.26	16.5	2.11	11.2	1.91
		Mixed mode	4.4	0.21	15.1	3.03	10.7	1.87
	No.2	Mode I	6.6	0.33	15.1	2.91	13.1	1.61
		Quasi mode II	4.8	0.51	15.7	1.99	14.4	2.19
		Mixed mode	5.9	0.29	14.8	3.66	13.3	2.87
	No.3	Mode I	7.3	0.12	14.6	2.67	14.7	1.82
		Quasi mode II	6.8	0.46	15.1	1.61	16.6	1.66
		Mixed mode	6.9	0.15	14.1	2.22	15.8	2.71
10	No.1	Mode I	0.2	--	23.4	3.07	12.0	1.26
		Quasi mode II	0.2		23.9	1.89	12.4	1.06
		Mixed mode	0.3		24.7	5.15	11.1	1.88
	No.2	Mode I	0.1		21.3	2.17	16.4	1.78
		Mode II	0.5		22.7	3.93	18.7	3.71
		Mixed mode	0.2		23.0	4.66	16.5	2.62
	No.3	Mode I	0.7		20.9	4.00	18.7	2.33
		Quasi mode II	0.6		21.6	3.72	22.7	4.10
		Mixed mode	0.1		22.1	2.24	19.2	3.82

**Table 6 materials-12-02675-t006:** Modeling process of the SCB numerical model.

Modeling Step	Modeling Procedure
Step One: built the numerical sample without aggregate, mastic and void	(a) Set test sample size (b) Create uniformly sized balls
Step Two: built the numerical sample without void	(a) Set gradation and asphalt mastic content (b) Randomly create polygon characteristics for coarse aggregates (c) Create ball clusters in the numerical sample prepared in step one according to (b)
Step Three: finalize the numerical sample	(a) Set porosity (b) Randomly create void distribution and size (c) Delete balls in the numerical sample prepared in step two according to (b)
Step Four: Give the contact models to the corresponding contacts in the numerical sample prepared in Step Three	

**Table 7 materials-12-02675-t007:** Errors between the laboratory data and simulation results.

*T* (°C)	Sample Type	Fracturing Mode	Average Error (%)				
			Fracture Toughness	Crack Extended Time	Crack Length in Aggregate	Crack Length in Mastic	Crack Length in Interface
−6	No.1	Mode I	1.17	2.61	1.54	2.91	2.98
		Quasi mode II	2.26	3.27	2.57	3.20	1.14
		Mixed mode	2.54	2.01	1.42	2.19	3.63
	No.2	Mode I	3.63	2.80	2.68	3.71	3.94
		Quasi mode II	3.39	4.17	5.61	5.71	3.05
		Mixed mode	2.40	6.04	2.67	4.67	2.49
	No.3	Mode I	4.81	2.73	3.61	4.19	4.55
		Quasi mode II	3.19	4.44	2.40	3.89	3.57
		Mixed mode	2.93	3.69	3.58	4.73	3.01
10	No.1	Mode I	1.89	2.81	--	3.24	3.15
		Quasi mode II	2.15	3.23		4.84	3.97
		Mixed mode	2.00	3.19		4.26	3.12
	No.2	Mode I	3.66	3.97		2.69	3.13
		Mode II	4.93	4.02		6.16	4.90
		Mixed mode	3.50	5.38		3.48	6.66
	No.3	Mode I	4.35	4.02		4.05	3.98
		Quasi mode II	4.62	5.68		4.30	4.69
		Mixed mode	3.91	4.45		3.55	4.07

**Table 8 materials-12-02675-t008:** Coarse aggregate characteristics in different DEM models.

Model	Zone I			Zone II			Other Zone		
	Orientation	Flatness	Content	Orientation	Flatness	Content	Orientation	Flatness	Content
0° model	0°	1.0	No.3	0°	1.0	No.3	0°	1.0	No.3
45° model	45°			45°			45°		
90° model	90°			90°			90°		
1.0 model	0°	1.0	No.3	0°	1.0	No.3	0°	1.0	No.3
1.1 model		1.1			1.1			1.1	
1.2 model		1.2			1.2			1.2	
1# model	Random	Random	No.3	Random	Random	No.2	Random	Random	No.3
2# model			No.3			No.3			
3# model			No.2			No.3			

**Table 9 materials-12-02675-t009:** Performance ratio of the fracture toughness.

Temperature	Fracture Loading Mode	*F*_0°_/*F*_45°_	*F*_0°_/*F*_90°_	*F*_1.0_/*F*_1.1_	*F*_1.0_/*F*_1.2_	*F*_1#_/*F*_2#_	*F*_1#_/*F*_3#_
−6 °C	Mode I	1.03	1.06	1.03	1.11	0.99	1.04
	Quasi-Mode II	1.01	1.04	1.05	1.12	0.97	1.04
	Mixed-Mode I and II	1.01	1.05	1.02	1.13	0.97	1.04
10 °C	Mode I	1.05	1.09	1.14	1.32	0.97	1.10
	Quasi mode II	1.04	1.10	1.16	1.24	0.92	1.14
	Mixed mode I&II	1.06	1.12	1.15	1.28	0.93	1.15

**Table 10 materials-12-02675-t010:** Performance ratio of the time at which the peak load occurs.

Temperature	Fracture Loading Mode	*C*_0°_/*C*_45°_	*C*_0°_/*C*_90°_	*C*_1.0_/*C*_1.1_	*C*_1.0_/*C*_1.2_	*C*_1#_/*C*_2#_	*C*_1#_/*C*_3#_
−6 °C	Mode I	1.02	1.01	1.01	1.03	1.01	1.02
	Quasi mode II	1.02	1.02	1.02	1.04	0.96	1.00
	Mixed mode I&II	1.01	1.03	1.02	1.05	0.97	0.98
10 °C	Mode I	1.02	1.05	1.04	1.08	0.98	1.05
	Quasi mode II	1.03	1.07	1.05	1.15	0.94	1.09
	Mixed mode I&II	1.03	1.06	1.05	1.13	0.95	1.06

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
