# Peer review of "Investigating the Effect of Aggregate Characteristics on the Macroscopic and Microscopic Fracture Mechanisms of Asphalt Concrete at Low-Temperature"

_materials, 2019, doi:10.3390/ma12172675_

Round 1

Reviewer 1 Report

·     The article still needs some grammatical and syntax improvements. Use of English service center is recommended. A few examples of the English errors are as follows (These are just a few examples):

o   The fracture toughness and the crack extended time were reduced with the increased aggregate orientation and flatness and the decreased aggregate content

o   he number of failed contacts in the interface in quasi mode II was slightly higher than that in other fracturing modes

o   And the aggregate distribution in the crack initiation zone greatly influenced the crack resistance, particularly at 10°C.

o   required a high cost and a lot of time on manufacturing samples

o   they were difficult to clearly reveal the effect of some important aggregate characteristics

o   The above studies contributed considerably to our understanding of cracking mechanisms of asphalt concrete.

o   The asphalt mastic was obtained by mixing the asphalt binder and the aggregates whose diameter were smaller than 1.18mm

o    which is due to that the aggregate crack resistance is reduced because the aggregates grow thinner

o   It was difficult to keep the void distribution constant in different aggregate distribution

·       The authors should justify why they chose two temperatures rather than a range including at least 5 different temperatures. Many conclusions could not be very reliable with only two temperature data. If possible, adding data for another temperature is recommended.

·     Overall the authors are strongly recommended to establish the reasons of the study and what the possible areas are that this study is trying to clarify. At this point the reason for which this study is conducted for is vague and undetermined. For example, the modes shown in Figure 4 are important?

·         The authors are encouraged to elaborate on the asphalt mastic and asphalt binder describing Table 3. It should be said that the asphalt mastic contains binder as well.

·         More details on Table 3 are required.

·         The authors are encouraged to search thoroughly for “Temperature effects on the behavior of grouts” and “Behavior of Concrete under Environmental Effects”. There are many recent works similar to this subject which could be included in introduction.

·         Introduction needs to include more works, and the introduction elaborates on the subject sporadically.

·       A few people might through know about SCB. It is recommended to add some information about the procedures, instruction and reasons of implementation of this method.

·       Figure 4 should be revised and checked.

·      The authors mentioned that “the crack extended time are increased with the increasing course aggregate content, implying that the decrease of coarse aggregate content brings a negative effect on the fracture resistance of asphalt concrete.” previous works indicated that there is optimal percentage of the aggregate after which if the aggregate content is decreased, then the fracture resistance is negatively affected. The reason that author reached to this kind of conclusions is due to lack of extended data, and only having a few samples.

·      What is the reason that “away from the main contacts ” have higher chance of F contacts at temperature 10c for all the modes compared to temperature -6c.

·         Figure 14 needs more details on the figure.

·        Figure 15 d . seems to have a significantly different crack initiation which could be due to many issues. What is the reason for such behavior?

·      There are many statements in the article which are merely explanation of the figures without any details about the reasons of such observations. The author should make deeper conclusions rather than mere restatement of the figures.

o   At 10°C, the number of failed contacts has a greater variation with the change of aggregate flatness, of which the effect trend increases gradually with the increased flatness.

o   It should be explained that most of the failed contacts occurred in the interface and away from the main crack path are invisible failed contacts.

Author Response

Dear Reviewer:

Thank you for your comments concerning our manuscript entitled “Investigating the Effect of Aggregate Characteristics on the Macroscopic and Microscopic Fracture Mechanisms of Asphalt Concrete at Low-temperature” (ID: materials-561164).Those comments are all valuable and very helpful for revising and improving our paper, as well as the important guiding significance to our researches. We have revised the manuscript according to your detailed suggestions. The main corrections in the paper and the responds to the reviewer’s comments are as flowing: 

(1) The article still needs some grammatical and syntax improvements. Use of English service center is recommended.

Authors’ Reply:

The language has been revised with the help of the English Editing Service provided by MDPI. Attached please find the editorial certificate.

(2) The authors should justify why they chose two temperatures rather than a range including at least 5 different temperatures. Many conclusions could not be very reliable with only two temperature data. If possible, adding data for another temperature is recommended.

Authors’ Reply:

The reviewer is right. The temperature and environmental effects are important for the fracture behavior of asphalt concrete. According to our literature review, there are few scarce studies on the composite analysis of mixed fracturing modes, aggregate characteristics and temperatures on the cracking characteristics of asphalt concrete. However, it is an enormous and complex work to clearly reveal the above relationship. In this study, the main innovative contribution is to investigate the coupling effect of mixed fracturing modes and aggregate characteristics on the cracking behavior of asphalt concrete at low-temperature. The effect of temperatures will be systematically drawn in our future research.

Moreover, according to the studies of Pirmohammad et al. (2014) and Yin et al. (2015), the low-temperature cracks of asphalt concrete present the elastic-brittle characteristics when the temperature is lower than 0°C, and that will be significantly changed when the temperature is higher than 5°C because the effect of viscoelasticity provided by asphalt mastic is gradually increased. Hence, according to the previous studies and our equipment conditions, two typical temperatures (-6°C and 10°C) are selected in this study to investigate the different crack propagation modes of asphalt concrete.

The above has been supplemented in the revised manuscript (Line 100~109 and 453~455).

Pirmohammad, S., and Ayatollahi, M., R. (2014). “Fracture resistance of asphalt concrete under different loading modes and temperature conditions”. Construction and Building Materials, 53(4), 235-242.

Yin, A., Yang, X., Zeng, G., and Gao, H. (2015). “Experimental and numerical investigation of fracture behavior of asphalt mixture under direct shear loading”. Construction and Building Materials, 86, 21-32.

(3) Overall the authors are strongly recommended to establish the reasons of the study and what the possible areas are that this study is trying to clarify. At this point the reason for which this study is conducted for is vague and undetermined. For example, the modes shown in Figure 4 are important?

Authors’ Reply:

The research significance of this study has been explained in the revised manuscript (Line 35~54).

(4) The authors are encouraged to elaborate on the asphalt mastic and asphalt binder describing Table 3. It should be said that the asphalt mastic contains binder as well.

Authors’ Reply:

The relationship between asphalt mastic and asphalt binder has been elaborated in the revised manuscript (Line 120~127 and Figure 1).

(5) More details on Table 3 are required.

Authors’ Reply:

Table 3 has been revised.

(6) The authors are encouraged to search thoroughly for “Temperature effects on the behavior of grouts” and “Behavior of Concrete under Environmental Effects”. There are many recent works similar to this subject which could be included in introduction.

Authors’ Reply:

The reviewer is right. The temperature effect is very important.

We want to explain that asphalt concrete and cement concrete are the two different materials that have significant differences. Hence, in the revised manuscript (Line 60~62, 76~79, 86~88 and 541~556), only the literatures about asphalt concrete are supplemented in the section “Introduction” according to reviewer’s suggestion.

(7) Introduction needs to include more works, and the introduction elaborates on the subject sporadically.

Authors’ Reply:

The section “Introduction” has been improved in the revised manuscript (Line 35~109).

(8) A few people might through know about SCB. It is recommended to add some information about the procedures, instruction and reasons of implementation of this method.

Authors’ Reply:

The SCB test has been introduced more detailedly in the revised manuscript (Line 144~165 and Figure 2~5).

(9) Figure 4 should be revised and checked.

Authors’ Reply:

Figure 4 has been revised (Please see Figure 6 in the revised manuscript). We select the loading conditions of SCB tests according to the study of Ameri et al. (2012).

(10) The authors mentioned that “the crack extended time are increased with the increasing course aggregate content, implying that the decrease of coarse aggregate content brings a negative effect on the fracture resistance of asphalt concrete.” previous works indicated that there is optimal percentage of the aggregate after which if the aggregate content is decreased, then the fracture resistance is negatively affected. The reason that author reached to this kind of conclusions is due to lack of extended data, and only having a few samples.

Authors’ Reply:

The reviewer is right. The analysis is not rigorous. The relevant content has been corrected in the revised manuscript (Line 193~196 and 560~564).

(11) What is the reason that “away from the main contacts” have higher chance of failed contacts at temperature 10c for all the modes compared to temperature -6c.

Authors’ Reply:

This is because that the difference of aggregate stiffness and mastic stiffness increases with the rising temperature. At -6°C, the difference of aggregate stiffness and mastic stiffness is relatively small. Then the cracks are intended to pass through the aggregates and asphalt mastics directly. The number of failed contacts away from the main crack path is very limited. At 10°C, the cracks are intended to pass through the asphalt mastic and aggregate-mastic interface and bypass the aggregates because of the increased stiffness difference between aggregates and mastics. Due to the complexity of the aggregate distribution, the cracks are more complex, and the failed contacts are easy to appear away from the main crack path. The fracturing modes can influence the distribution of the failed contacts, but the effect is weaker than that of the temperatures (from -6°C to 10°C).

The above has been supplemented in the revised manuscript (Line 347~356).

(12) Figure 14 needs more details on the figure.

Authors’ Reply:

Figure 14 has been revised (Please see Figure 17 in the revised manuscript).

(13) Figure 15 d. It seems to have a significantly different crack initiation which could be due to many issues. What is the reason for such behavior?

Authors’ Reply:

The reviewer may refer to 1# model in mixed mode I&II at 10°C that is cracked from the right support. It is a special case which can be observed in a very few of numerical simulations and laboratory tests. The sample with this phenomenon both in numerical simulations and laboratory tests is considered as the invalid sample and must be retested. However, in order to show and explain this phenomenon, we especially select the numerical sample with this phenomenon to display in Figure 15.

The phenomenon is due to that the materials near the right support are small aggregates and asphalt mastics. The crack resistance of these small aggregates and asphalt mastics near the right support is much lower than that of the large aggregates in zone I near the pre-cut notch. Hence the crack is inclined to appear near the right support. This phenomenon also can be observed in a few experiments at 10°C accidentally, of which the main reason is the difference of aggregate distribution between near the pre-cut notch and the support.

The above has been supplemented in the revised manuscript (Line 418~426).

(14) There are many statements in the article which are merely explanation of the figures without any details about the reasons of such observations. The author should make deeper conclusions rather than mere restatement of the figures. (i) At 10°C, the number of failed contacts has a greater variation with the change of aggregate flatness, of which the effect trend increases gradually with the increased flatness. (ii) It should be explained that most of the failed contacts occurred in the interface and away from the main crack path are invisible failed contacts.

Authors’ Reply:

The detailed analysis has been supplemented in the revised manuscript (Line 357~361 and 372~376).

We have checked the manuscript and revised it in detail according to your comments. We submit here the revised manuscript as well as a list of changes, and some minor errors have been modified in the paper which does not list above.

We would like to express our great appreciation to you for comments on our paper. If you have any question about this paper, please don’t hesitate to let me know.

Yours sincerely,

Xue Jinshun

Co-author: Yinshan Xu, Email: [email protected];

Yingjun Jiang, Email: [email protected];

Jinshun Xue, Email: [email protected];

Jiaolong Ren, Email: [email protected]

Reviewer 2 Report

The masuscript deals with a subject of current relevance clearkly falling within the scope of the journal. In tis reviewer's opinion the paper deserves to be accepted for publication, as it presents fresh experimental results and interesting numerical elaboration.

A Minor revision of the current submission is requested to the Authors, who can refer to the comments reported throughout the marked manuscript attached at the present report.

Author Response

Dear Reviewer:

Thank you for your comments concerning our manuscript entitled “Investigating the Effect of Aggregate Characteristics on the Macroscopic and Microscopic Fracture Mechanisms of Asphalt Concrete at Low-temperature” (ID: materials-561164).Those comments are all valuable and very helpful for revising and improving our paper, as well as the important guiding significance to our researches. We have revised the manuscript according to your kind advices. The main corrections in the paper and the responds to the reviewer’s comments are as flowing:

The manuscript deals with a subject of current relevance clearly falling within the scope of the journal. In the reviewer's opinion the paper deserves to be accepted for publication, as it presents fresh experimental results and interesting numerical elaboration. A Minor revision of the current submission is requested to the Authors, who can refer to the comments reported throughout the marked manuscript attached at the present report.

(1) Abstract should be "sharper" and more concise. Five sentences should mention scope, motivation, methodology, results and expected impact of the research.

Authors’ Reply:

The abstract has been revised (Line 16~30).

(2) Please, revise English in section 1.

Authors’ Reply:

The language has been revised with the help of the English Editing Service provided by MDPI. Attached please find the editorial certificate.

(3) Please, rearrange the tables (Table 4) so that they fit in the pages.

Authors’ Reply:

The format has been revised according to the MDPI standard.

(4) This section (Conclusion) should be completely rewritten. After an introductory sentence, the main findings have to be listed in concise items. Please, do not refer to mere empirical evidence, but try to point out the mechanical reasons behind them.

Authors’ Reply:

The conclusion has been rewritten in the revised manuscript (Line 428~455).

We have checked the manuscript and revised it in detail according to your comments. We submit here the revised manuscript as well as a list of changes, and some minor errors have been modified in the paper which does not list above.

We would like to express our great appreciation to you for comments on our paper. If you have any question about this paper, please don’t hesitate to let me know.

Yours sincerely,

Xue Jinshun

Co-author: Yinshan Xu, Email: [email protected];

Yingjun Jiang, Email: [email protected];

Jinshun Xue, Email: [email protected];

Jiaolong Ren, Email: [email protected]

Reviewer 3 Report

The authors present interesting work on “the effect of aggregate characteristics on the macroscopic and microscopic fracture mechanisms of asphalt concrete at low-temperature”.

Overall the document is well written, the experimental campaign is adequate and the results analysis well structured. However, before being considered for publication this paper should have the following corrections:

- At the beginning of the paper it is essential to emphasize two things::

1. A good survey of the current state of knowledge is essential. The authors present only a few references to the discrete element modelling; not enough! It is necessary to understand what is already known about macroscopic and microscopic fracture mechanisms of asphalt concrete at low temperature.

2. It is essential that the authors present the novelty of this study, demonstrating its contribution to the current state of knowledge on the subject.

- Only 3 mixes produced? They must adequately justify the application of this numerical method with only 3 mixtures!

- The application of the discrete element modeling should be introduced and explained in more detail.

- The application and development of the method is interesting, however, the results presented should be crossed with results of other authors obtained identically or by another method. Benchmarking is essential!

- The conclusions could be more assertive about the initial objective of the paper. In practice we want to know and actually bluntly what is"the effect of aggregate characteristics on the macroscopic and microscopic fracture mechanisms of asphalt concrete at low-temperature”.

Author Response

Dear Reviewer:

Thank you for your comments concerning our manuscript entitled “Investigating the Effect of Aggregate Characteristics on the Macroscopic and Microscopic Fracture Mechanisms of Asphalt Concrete at Low-temperature” (ID: materials-561164).Those comments are all valuable and very helpful for revising and improving our paper, as well as the important guiding significance to our researches. We have revised the manuscript according to your kind and detailed suggestions. The main corrections in the paper and the responds to the reviewer’s comments are as flowing:

The authors present interesting work on “the effect of aggregate characteristics on the macroscopic and microscopic fracture mechanisms of asphalt concrete at low-temperature”. Overall the document is well written, the experimental campaign is adequate and the results analysis well structured. However, before being considered for publication this paper should have the following corrections:

At the beginning of the paper it is essential to emphasize two things:

(1) A good survey of the current state of knowledge is essential. The authors present only a few references to the discrete element modelling; not enough! It is necessary to understand what is already known about macroscopic and microscopic fracture mechanisms of asphalt concrete at low temperature.

Authors’ Reply:

The relevant content has been supplemented in the revised manuscript (Line 69~99).

(2) It is essential that the authors present the novelty of this study, demonstrating its contribution to the current state of knowledge on the subject.

Authors’ Reply:

The innovation of this study has been explained more clearly in the revised manuscript (Line 103~105).

(3) Only 3 mixes produced? They must adequately justify the application of this numerical method with only 3 mixtures!

Authors’ Reply:

In this study, to ensure the reliability of the numerical simulations, the parallel testing is implemented in numerical simulations just as that used in experimental tests. In experimental parallel tests, asphalt concrete samples usually have the same raw materials and material composition and the random material distribution. Accordingly, in numerical parallel tests, numerical models have the same material parameters and material composition and the random material distribution. As previously mentioned, for each loading condition and concrete composition, six samples were tested in experimental tests. Similarly, six numerical models were simulated successfully for each loading condition and concrete composition.

The above has been supplemented in the revised manuscript (Line 231~238).

(4) The application of the discrete element modeling should be introduced and explained in more detail.

Authors’ Reply:

The relevant content has been supplemented in the revised manuscript (Line 212~224 and Figure 8).

(5) The application and development of the method is interesting, however, the results presented should be crossed with results of other authors obtained identically or by another method. Benchmarking is essential!

Authors’ Reply:

We have supplemented some analysis of other authors that have connections with our studies in the revised manuscript (Line 342~344 and 380~384).

(6) The conclusions could be more assertive about the initial objective of the paper. In practice we want to know and actually bluntly what is "the effect of aggregate characteristics on the macroscopic and microscopic fracture mechanisms of asphalt concrete at low-temperature”.

Authors’ Reply:

The conclusion has been rewritten in the revised manuscript (Line 428~455).

We have checked the manuscript and revised it in detail according to your comments. We submit here the revised manuscript as well as a list of changes, and some minor errors have been modified in the paper which does not list above.

We would like to express our great appreciation to you for comments on our paper. If you have any question about this paper, please don’t hesitate to let me know.

Yours sincerely,

Xue Jinshun

Co-author: Yinshan Xu, Email: [email protected];

Yingjun Jiang, Email: [email protected];

Jinshun Xue, Email: [email protected];

Jiaolong Ren, Email: [email protected]

Round 2

Reviewer 1 Report

Although the authors mentioned about why just two temperatures are chosen, there is still low reliability of the results. To understand the trends and have the basic predictions of the behavior, at least three temperature s are needed.

There are still many statements in the article which are merely explanation of the figures. The author should make deeper conclusions rather than mere restatement of the figures.

More similar works are as follows, the authors are welcome to consider these works upon their discretion as well.

Farzampour, Alireza. "Compressive Behavior of Concrete under Environmental Effects." IntechOpen, 2019.

Thanks.

Author Response

Response to Reviewers’ Comments

Dear Reviewers:

On behalf of my co-authors, we thank you very much for giving us an opportunity to revise our manuscript again, we appreciate reviewer very much for your positive and constructive comments and suggestions on our manuscript entitled “Investigating the Effect of Aggregate Characteristics on the Macroscopic and Microscopic Fracture Mechanisms of Asphalt Concrete at Low-temperature” (ID: materials-561164).

We have studied the comments carefully and have made revision in the paper. We have tried our best to revise our manuscript according to your comments. We took into account of your comment, the main corrections in the paper and the responds to the comments are as flowing:

Reviewer 1:

(1) Although the authors mentioned about why just two temperatures are chosen, there is still low reliability of the results. To understand the trends and have the basic predictions of the behavior, at least three temperature s are needed.

Authors’ Reply:

The experimental results at 0°C have been supplemented to further describe the effect of temperature on the fracture behaviour of asphalt concrete in the revised manuscript (Table 4~5 and Line 210~215).

(2) There are still many statements in the article which are merely explanation of the figures. The author should make deeper conclusions rather than mere restatement of the figures.

Authors’ Reply:

The relevant contents have been supplemented in the revised manuscript (Line 318~319, 321, 325 and 373~374).

(3) More similar works are as follows, the authors are welcome to consider these works upon their discretion as well.

Authors’ Reply:

The literatures suggested by the reviewer have been supplemented in the revised manuscript (Line 61~62 and 574~577).

Reviewer 2 Report

 The authors duly revised their original submission and, hence, the current version of the manuscript can be accepted for publication.

Reviewer 3 Report

The initial version has been changed as requested.

Under these conditions I am of the opinion that the paper can be considered for publication.

Author Response

Response to Reviewers’ Comments

Dear Reviewer:

On behalf of my co-authors, we thank you very much for giving us an positive and constructive comments and suggestions on our manuscript entitled “Investigating the Effect of Aggregate Characteristics on the Macroscopic and Microscopic Fracture Mechanisms of Asphalt Concrete at Low-temperature” (ID: materials-561164).

Reviewer 2:

The initial version has been changed as requested. Under these conditions I am of the opinion that the paper can be considered for publication.

Authors’ Reply:

Thank you very much for your kind advice.

Round 3

Reviewer 1 Report

The authors answered the issues.